# Evaluation of Rock Brittleness Index under Dynamic Load

Diyuan Li *, Minggang Han and Quanqi Zhu

School of Resources and Safety Engineering, Central South University, Changsha 410083, China; 205511011@csu.edu.cn (M.H.); quanqi_zhu@csu.edu.cn (Q.Z.)
* Correspondence: diyuan.li@csu.edu.cn

**Abstract:** Rock is a typical brittle material, and the evaluation of its brittleness index has important guiding significance for hard rock resource exploitation, unconventional oil and gas resource exploitation, mechanical driving efficiency, rock burst prediction, and dynamic disaster prevention and control. At present, brittleness index often measures the brittleness of rock under static load; thus, whether it is applicable to dynamic load is worth exploring. In this study, static and dynamic uniaxial compression tests and Brazilian splitting tests were carried out on five kinds of rocks, including fine granite, coarse granite, shale, marble, and sandstone, using the INSTRON−1346 test system and split−Hopkinson pressure bar (SHPB), respectively. The brittleness index values of different rocks under static and dynamic load were determined, and the changes in the brittleness of rocks under different loading methods and different strain rates were studied. The definition of brittleness and the applicability of existing brittleness indices were also discussed. It was found that the loading rate amplified the variation of the brittleness characteristics of rock. When static load changes to dynamic load, the brittleness of rocks increases, and the brittleness relationship between different rocks remains unchanged. The more brittle the rock is under static load, the greater the range of brittleness enhancement is under dynamic load. It was also found that the brittleness of sandstone had an obvious effect on the strain rate. The brittleness of rock increases with the increase in strain rate, and the greater the strain rate, the greater the brittleness enhancement degree. These research results can provide reference values for dynamic disaster prevention and safe construction of deep rock projects such as mines and tunnels.

**Keywords:** rock mechanics; brittleness index; SHPB; total stress–strain curve

## 1. Introduction

Rock is a natural material, composed of various minerals and formed through long-term geological processes. The differences in properties among its components and internal microfissures determine the mesoscopic heterogeneity of rock, which is the internal basis for the differences in the failure of different types of rock. Plasticity usually means that the material can maintain a complete and continuous deformation property, while brittle failure means that the material loses continuity and the initiation and expansion of cracks occurs. The brittleness characteristics of rock have important guiding significance for hard rock resource exploitation, rock burst prediction, mechanical excavation efficiency, and unconventional oil and gas resource exploitation [1–7]. Therefore, the correct evaluation of rock brittleness is crucial for the safe design and selection of mining machinery, effective reconstruction of unconventional shale gas reservoirs, and stability evaluation of the surrounding rock [8].

For the definition of brittleness, researchers mainly begin with the characteristics of brittle materials. Morley (1944) [9] and Hetényi (1950) [10] defined brittleness as a lack of ductility. Howell (1960) [11] defined brittleness as a behavior that produces little or no plastic strain during rock failure or fracture. Obert and Duvall (1967) [12] defined brittleness as the occurrence of fracture failure at or just above the yield stress. Bates and Jackson [13] (1984) believed that brittleness was an inherent property of materials

which was characterized by the fracturing of materials with little or no plastic deformation. They interpreted fractured rocks with deformation or strain less than 3–5% as brittle rocks. Andreev [14] (1995), Yilmaz [15] (2009), Mogi [16] (1966), and Hoek and Brown et al. [17] (1980) agreed that brittleness should be defined as the fracture ability of a material without obvious permanent deformation in tensile or compression tests. In addition, Honda and Sanada [18] (1956) used fissure formation capacity to reflect rock brittleness. Lawn and Marshall [19] (1979) defined brittleness as the ease of crack propagation. Allaby [20] (1991) connected the brittleness of rock with the loss of cohesion in the elastic range of rock under load. Brittleness was considered to be the ability of rocks of a certain strength to lose cohesion along certain surfaces when the stress exceeds the elastic limit. Dollinger [21] (1998) defined brittleness as the ease with which rocks form debris during the indentation process. Lawn and Marshall [22] (1979) referred to brittleness as one of the most elusive mechanical properties, measuring the competition between deformation (residual or plastic indentation) and fracture under indentation.

In order to quantify the changes in rock brittleness, domestic and foreign scholars put forward the concept of a brittleness index. However, due to the inconsistent definition of brittleness and the different application fields, there is no internationally recognized standard for the determination of a rock brittleness index [7]. Some brittleness indices are shown in Table 1. Based on the relationship between the tensile and compressive strength of rocks and brittleness, Hucka and Das [22] proposed the classical brittleness index, $B_1$. Altindag [23,24] studied the uniaxial tension−uniaxial compressive strength curve and found that the area surrounded by the curve could be used to quantify the brittleness of rock, so he proposed a brittleness index, $B_2$, based on the uniaxial tension−uniaxial compressive strength curve. Based on the strain characteristics of brittle rocks, Hucka and Das [22] proposed that the ratio of elastic strain to total strain be used as a brittleness index, $B_3$, to measure the brittleness of rocks. Based on the post−peak failure characteristics of brittle rocks, Meng and Zhou [8,25] considered the relative magnitude and absolute rate of post−peak stress drop in stress–strain curves, and proposed brittleness indices $B_4$ and $B_5$. Based on the total stress–strain curve, Xia [26] proposed a brittleness index, $B_6$, based on stress reduction rate and elastic energy. Based on the energy relationship, Hucka and Das [22] proposed the brittleness index $B_7$, which measured rock brittleness by the ratio of elastic energy to total energy at failure. Tarasov and Potvin [27] proposed brittleness indices $B_8$ and $B_9$ under two different failure modes, considering the energy balance relationship in the post−failure region of the rock. Ai [28] defined brittleness as the ability of rock to accumulate elastic energy in the pre−peak stage and self−sustaining fracture propagation in post−peak stage, and proposed brittleness indices $B_{10}$ and $B_{11}$. The above brittleness indices indican be obtained by conventional mechanical tests (mainly uniaxial compression tests). In addition, the brittleness index of rock also includes a brittleness index based on mineral composition, a brittleness index based on elastic parameters, etc., which will not be discussed here.

**Table 1.** List of common brittleness indices.

| Brittleness Index | Parameter | Source |
|---|---|---|
| $B_1 = \sigma_c / \sigma_t$ <br> $B_2 = \sqrt{\sigma_c \sigma_t / 2}$ | $\sigma_c$ is uniaxial compressive strength, $\sigma_t$ is uniaxial tensile strength | Hucka and Das (1974) [22] <br> Altindag (2010) [23,24] |
| $B_3 = \varepsilon_e / \varepsilon_t$ | $\varepsilon_e$ is the recoverable strain after loading, $\varepsilon_t$ is the total strain of loading | Hucka and Das (1974) [22] |
| $B_4 = \left[ (\sigma_p - \sigma_r) / \sigma_p \right] \times [\lg|k_{ac}| / 10]$ <br><br> $B_5 = (\sigma_p - \sigma_r) \times [\lg|k_{ac}| / 10]$ | $\sigma_p$ is the peak intensity, $\sigma_r$ is the residual strength <br> $k_{ac}$ is the slope of a straight line from the peak strength point to the starting point of residual strength | Meng FZ and Zhou H (2015) [8] <br><br> Meng FZ and Zhou H (2015) [8] |
| $B_6 = B_{POST} + B_E = \frac{\sigma_p - \sigma_r}{\varepsilon_r - \varepsilon_p} + \frac{(\sigma_p - \sigma_r)(\varepsilon_r - \varepsilon_p)}{\sigma_p \varepsilon_p}$ | $\varepsilon_p$ is the peak strain, $\varepsilon_r$ is the residual strain, $\varepsilon_i$ is the initiation strain, $\sigma_p$ is the peak stress, $\sigma_r$ the residual stress | Xia (2017) [26] |
| $B_7 = W_r / W_t$ | $W_r$ is recoverable strain energy, $W_t$ is the total strain energy | Hucka and Das (1974) [22] |
| $B_8 = (M - E) / M$ <br> $B_9 = E / M$ | $E$ is the pre−peak elastic modulus, and $M$ is the post−peak elastic modulus | Tarasov and Potvin (2013) [27] <br> Tarasov and Potvin (2013) [27] |
| $B_{10} = (U_r + U_d) / (U_e + U_d)$ <br> $B_{11} = U_a / (U_e + U_d)$ | $U_r$ is the fracture energy, $U_d$ is the dissipated energy, $U_e$ is the elastic strain energy, $U_a$ is the energy absorbed or released after the peak | Ai (2016) [28] <br> Ai (2016) [28] |

In practical engineering, rock mass is often subjected to dynamic load, and many researchers have carried out a significant number of studies on the relevant mechanical characteristics of rock under dynamic load [29–31]. However, studies on brittleness change and brittleness index tend to focus on static loading rather than dynamic loading. Xu [32] carried out dynamic compression tests on sandstone and analyzed the brittleness characteristics of coal measure sandstone under dynamic load. However, there has been no discussion regarding which brittleness index is suitable for measuring rock brittleness under dynamic load. Therefore, it is necessary to study the changes in the brittleness of different rocks under dynamic and static loads and whether the existing brittleness indices are suitable for measuring the brittleness of rocks under dynamic loads. In this paper, static and dynamic uniaxial compression tests and Brazilian splitting tests were carried out on five kinds of rocks, including fine granite, coarse granite, shale, marble, and sandstone, using the INSTRON−1346 test system and split−Hopkinson pressure bar (SHPB), respectively. The corresponding brittleness index values were determined by the dynamic stress–strain curve and compared with the brittleness index values under static load. The relationship between brittleness and strain rate was also studied by dynamic impact tests on sandstone under different strain rates. In addition, the effect of brittleness on macroscopic failure characteristics of rock was studied using a high−speed camera. The research results can provide reference values for dynamic disaster prevention and control, as well as the safe construction of deep rock projects such as mines and tunnels.

## 2. Materials and Methods

### 2.1. Test Equipment and Test Methods

The INSTRON 1346 testing system was used for the static test. It includes a control computer, loading system, pressure chamber, hydraulic transmission system, and data acquisition system. In the uniaxial compression test, in order to obtain the complete stress–strain curve of the rock, a load control with a loading rate of 90 kN/min was first adopted. When the loading reached about 75% of the uniaxial compressive strength, the loading control mode was changed to the displacement control with a loading rate of 0.12 mm/min until the specimen was damaged. In order to accurately measure the deformation of rock specimens, longitudinal and transverse strain gauges were pasted on two sides of the specimens to obtain their axial and circumferential strains. Using the sensor of the testing machine, the axial force and total displacement of the rock specimen were obtained in order to draw the complete stress–strain curve of the rock. In the Brazilian splitting test, the upper and lower steel bars were placed on the loading ends of the disc specimen, and the relative linear load was applied to cause them to fail along the diameter direction of the specimen. During the test, axial displacement was used to control the loading rate at 0.1 mm/min until the specimen failed. The results of the uniaxial compression tests and Brazilian splitting tests were processed according to the methods recommended by the International Society of Rock Mechanics (ISRM).

The SHPB test system was used for the dynamic test. The SHBP system consists of an incident rod, a transmission rod, an absorption rod, and a damping rod, all of which are made of high strength 40Cr alloy steel. Before the test, petroleum jelly was evenly applied onto the end faces of the specimen and bars to reduce the end friction effect [32]. A spindle−shaped bullet was used in the experiment to generate a half−sinusoidal load wave. The dynamic compression test and dynamic Brazilian splitting test were conducted on different rocks, and each rock was subjected to the same impact energy by controlling the pressure in the air chamber. In addition, dynamic compression tests and dynamic Brazilian splitting tests were carried out on sandstone at different strain rates by adjusting the pressure in the air chamber.

### 2.2. Test Specimen

Five rock materials, including fine granite, coarse granite, shale, marble, and sandstone, were selected as experimental research objects. All rocks had good integrity, uniform and

dense texture, and no defects visible to the naked eye. The rock specimens were prepared by drilling and cutting. The specimens were divided into three categories. The first was a standard cylindrical specimen with a diameter (D) of 50 mm and a height (H) of 100 mm. The second was a standard Brazilian split specimen with a diameter (D) of 50 mm and a height (H) of 25 mm. The third was the dynamic impact compression specimen with a diameter (D) of 50 mm and a height (H) of 50 mm. Non−parallel end faces and axial deviations were within the allowable limits of the International Society of Rock Mechanics (ISRM) standards. Figure 1 shows the specimens of five kinds of rocks after processing.

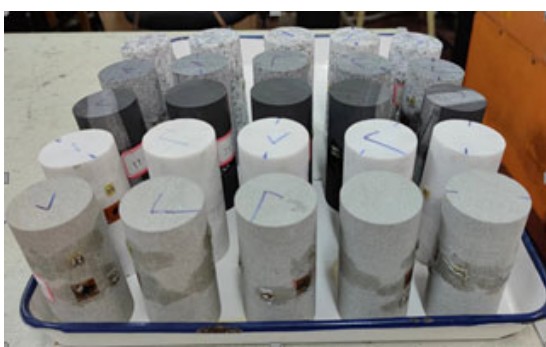
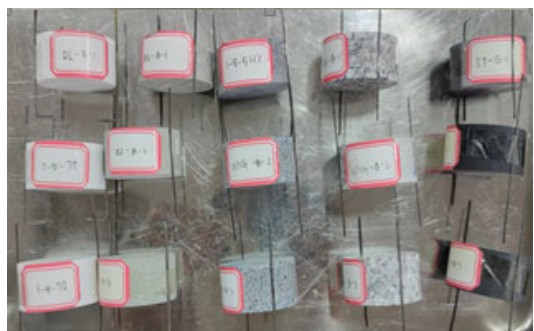
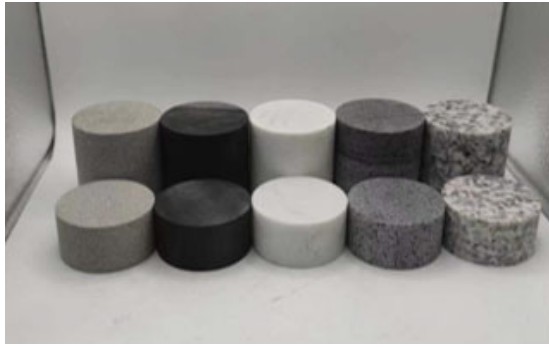
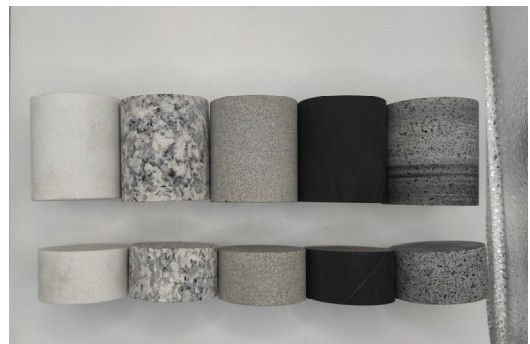

**Figure 1.** Five types of rock specimens.

In all tests, each rock type included three parallel specimens. The relevant parameters measured in parallel specimens were basically the same. The standard deviations of uniaxial compressive strength and dynamic compressive strength were below 5 MPa, the standard deviations of static Brazilian splitting strength were below 1 MPa, and the standard deviations of dynamic Brazilian splitting strength were below 3 MPa. Table 2 shows the average physical and mechanical parameters of different rocks; the average values of the parameters were taken from the three specimens.

**Table 2.** The average physical and mechanical parameters of rock specimens ($\sigma_{cs}$ is uniaxial compressive strength; $\sigma_{cd}$ is dynamic compressive strength; $\sigma_{ts}$ is static Brazilian splitting strength; $\sigma_{td}$ is dynamic Brazilian splitting strength; $E_S$ is the static elastic modulus; $E_D$ is the dynamic elastic modulus; $\dot{\varepsilon}_{cd}$ is the strain rate of the specimens in a dynamic compressive test).

| Rock Type | Density (g/cm³) | $\sigma_{cs}$ (MPa) | $\sigma_{cd}$ (MPa) | $\sigma_{ts}$ (MPa) | $\sigma_{td}$ (MPa) | $E_s$ (GPa) | $E_D$ GPa | $\dot{\varepsilon}_{cd}$ (s⁻¹) |
|---|---|---|---|---|---|---|---|---|
| Fine granite | 2.78 | 168.35 | 417.41 | 11.88 | 27.65 | 39.50 | 71.05 | 105 |
| Coarse granite | 2.64 | 137.70 | 306.84 | 7.42 | 15.25 | 31.07 | 41.37 | 104 |
| Shale | 2.37 | 117.17 | 209.55 | 5.32 | 8.72 | 24.84 | 25.19 | 90 |
| Marble | 2.83 | 101.37 | 190.31 | 4.28 | 7.81 | 35.10 | 38.52 | 103 |
| Sandstone | 2.32 | 79.36 | 154.72 | 4.80 | 8.13 | 14.19 | 18.50 | 95 |

### 2.3. Analysis of SHPB Dynamic Impact Test Data

The assumption that the rock sample is in the state of one−dimensional stress wave propagation and stress equilibrium was the prerequisite for an effective dynamic impact compression test. Based on the above two assumptions, the stress, strain, and strain rate of the specimen, along with the loading time, can be obtained using the following formula [33].

$$\begin{cases} \sigma(t) = \frac{AE}{2A_S}[\varepsilon_I(t) + \varepsilon_R(t) + \varepsilon_T(t)] \\ \varepsilon(t) = \frac{C}{l_S}\int_0^t [\varepsilon_I(t) - \varepsilon_R(t) - \varepsilon_T(t)]\mathrm{d}t \\ \dot{\varepsilon}(t) = \frac{C}{l_S}[\varepsilon_I(t) - \varepsilon_R(t) - \varepsilon_T(t)] \end{cases} \tag{1}$$

where $\sigma(t)$, $\varepsilon(t)$, and $\dot{\varepsilon}(t)$, respectively, represent the stress, strain, and strain rate of the specimen at a certain time. $E$, $C$, and $A$ are the elastic modulus, p−wave velocity, and cross−sectional area of the elastic bar, respectively. $l_S$ and $A_S$ are the length and cross−sectional area of the specimen. $\varepsilon_I(t)$, $\varepsilon_R(t)$, and $\varepsilon_T(t)$, respectively, represent the incident strain, reflected strain, and transmitted strain of the pressure bar at a certain time.

According to the stress waves collected in the impact test (including incident wave, reflected wave, and transmitted wave), the incident energy, reflected energy, and transmitted energy on the specimen were obtained. The calculation formula was as follows:

$$\begin{cases} E_I(t) = ECA\int_0^t \varepsilon_I^2(t)\mathrm{d}t \\ E_R(t) = ECA\int_0^t \varepsilon_R^2(t)\mathrm{d}t \\ E_T(t) = ECA\int_0^t \varepsilon_T^2(t)\mathrm{d}t \\ E_A(t) = E_I(t) - E_R(t) - E_T(t) \end{cases} \tag{2}$$

where $E_I(t)$, $E_R(t)$, $E_T(t)$ and $E_A(t)$ are incident energy, reflected energy, transmitted energy, and absorbed energy, respectively.

## 3. Results

### 3.1. Brittle Failure Characteristics of Rock under Static and Dynamic Loads

Figure 2 shows the stress–strain curves of rocks with different loading methods, and Figure 3 shows the changes in the compressive strength of five kinds of rocks. It can be seen that fine granite had the highest compressive strength under static load, followed by coarse granite, shale, marble, and sandstone. Under dynamic load, the compressive strength of the five kinds of rocks was greatly improved compared with that under static load.

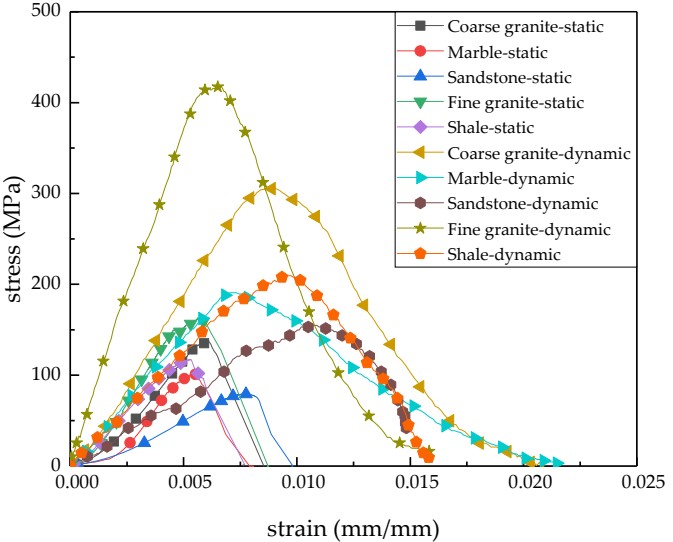

**Figure 2.** Stress–strain curves with different loading modes.

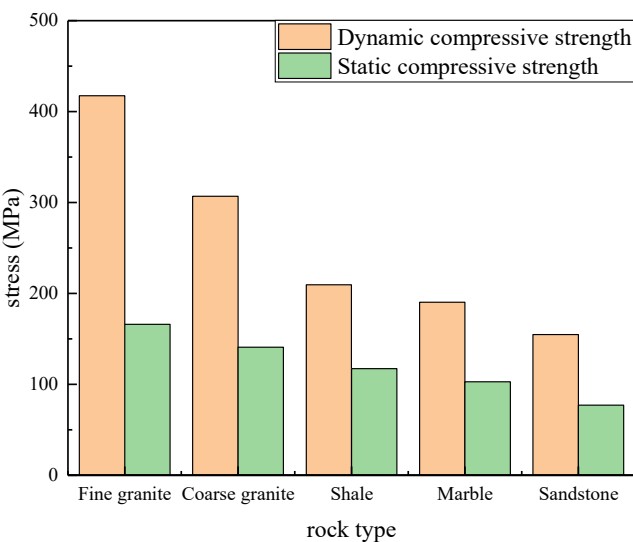

**Figure 3.** Change in compressive strength of rock samples.

Basu [34] conducted a large number of uniaxial compression tests on granite, schist, and sandstone, and divided the failure modes of all rock types into six types: axial splitting, shearing along single plane, double shear, multiple fracture, along foliation, and Y−shaped failure, as shown in Figure 4.The images of rock failure under different loading modes in this paper are shown in Figure 5. It can be seen that under static load, the failure mode of fine granite, marble, and sandstone was shearing along single plane, while the failure mode of coarse granite and shale was axial splitting. The failure modes of five kinds of rock under dynamic load were mainly axial splitting failure. Under the action of load, the elastic deformation of the rock was transformed into sudden fractures; this process is called brittle failure. In the test, none of the five kinds of rock showed obvious deformation after failure under different loading modes, demonstrating the characteristics of brittle failure.

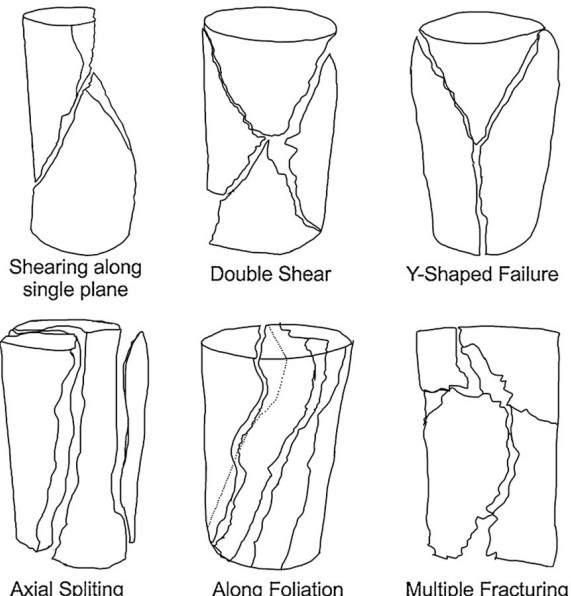

**Figure 4.** Schematic representation of different failure modes under uniaxial compression.

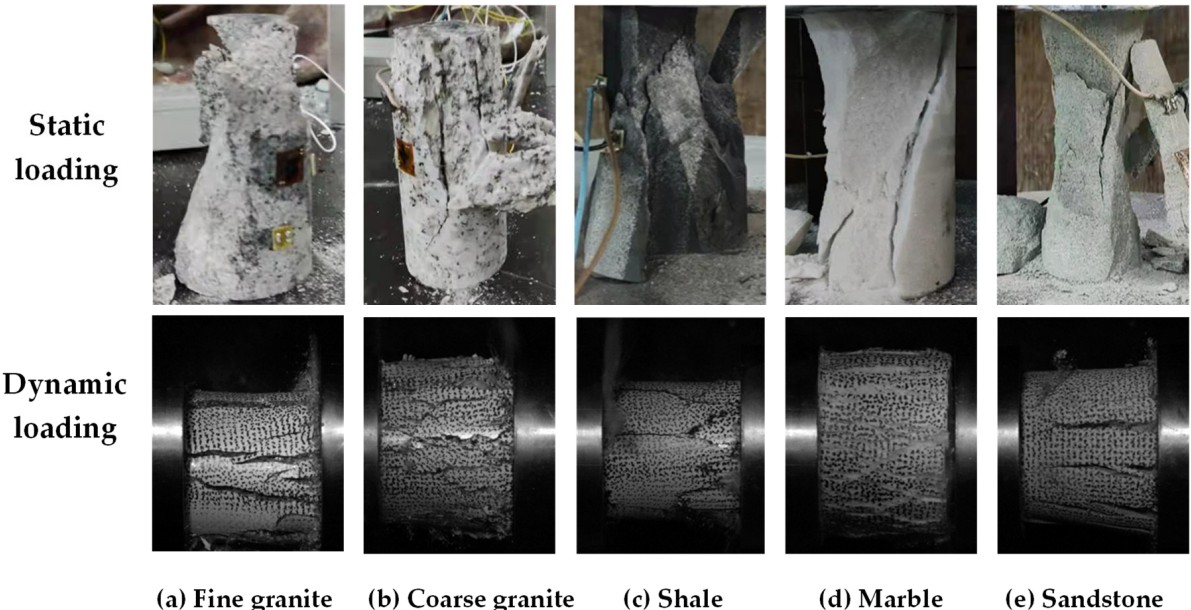

|  | (a) Fine granite | (b) Coarse granite | (c) Shale | (d) Marble | (e) Sandstone |

**Figure 5.** Rock failure under different loading modes.

Brittle failure is often the result of crack formation and development in rocks. Under static load, the loading rate is low and deformation occurs simultaneously, during which the whole rock specimen is compressed. At this time, the original microcracks in the specimen first close with the increase in pressure. Subsequently, when the compressive tensile stress exceeds the local tensile strength of the deficient tips, cracks are generated from these tips and propagate parallel to the direction of loading [35]. The crack propagation is affected by the microstructure of the specimen [36,37]. When the microstructure of the specimen does not hinder the crack propagation, the specimen is destroyed in axial splitting mode. However, the specimen microstructure restricts crack propagation, adjacent cracks, and close−range cracks generated by microcrack tips with appropriate orientation merge, releasing strain energy in the form of shear fracture and resulting in higher uniaxial compressive strength than that described in general axial splits [34]. In this study, fine granite had a higher uniaxial compressive strength than coarse granite, and the failure mode changed from axial splitting to shearing along a single plane. The uniaxial compressive strength of sandstone and marble was higher than that of similar rocks, so they also produced shear along a single plane, which is consistent with Basu's conclusion. In addition, when the specimen itself has a bedding structure, the crack propagation will be affected by the bedding structure direction due to the large difference in local tensile strength. The direction of shale bedding structure used in this paper is basically consistent with the direction of loading, so axial splitting failure occurs. Under dynamic load, the loading rate is high and the deformation of rock specimen propagates in the form of compression waves. Deformation occurs sequentially, and the specimen cannot be considered as a whole compression. Under the action of incident waves, cracks are generated along the loading direction of the rock. The crack propagation is affected by the compression wave, and the restriction of the microstructure on crack propagation becomes smaller, resulting in axial splitting failure.

*3.2. Applicability Analysis of Brittleness Index under Dynamic Load*

The rationality of the existing brittleness index to measure the brittleness of rock under static load has been verified by a significant number of tests, but its rationality for measuring the brittleness of rock under dynamic load still needs to be explored. According to the test results, the brittleness of rock under dynamic load was significantly improved compared with that under static load from the perspective of qualitative analysis. Therefore, an index that can quantitatively measure the brittleness of rock under dynamic load should have

the same law. Through the dynamic and static tests which we carried out, 11 brittleness indices of each specimen were calculated, and the average value of the parallel specimens as taken as the brittleness index value for each type of rock. The calculated results and the calculation results are shown in Table 3. Among these indices, the smaller the index values of $B_9$ and $B_{11}$, the stronger the brittleness; in contrast, the larger the index values of the other indices, the stronger the brittleness.

**Table 3.** Brittleness index values of different rocks under dynamic and static loading.

| Brittleness Index | Loading Mode | Brittleness Value of Rock | | | | |
| --- | --- | --- | --- | --- | --- | --- |
| | | Fine Granite | Coarse Granite | Shale | Marble | Sandstone |
| $B_1$ | Static | 13.982 | 18.978 | 22.039 | 23.996 | 16.068 |
| | dynamic | 15.096 | 20.118 | 24.044 | 24.361 | 19.031 |
| $B_2$ | Static | 31.400 | 22.868 | 17.648 | 14.834 | 13.596 |
| | dynamic | 75.967 | 48.373 | 30.218 | 27.264 | 25.079 |
| $B_3$ | Static | 0.753 | 0.723 | 0.794 | 0.566 | 0.722 |
| | dynamic | 0.910 | 0.852 | 0.883 | 0.735 | 0.785 |
| $B_4$ | Static | 0.477 | 0.472 | 0.471 | 0.467 | 0.466 |
| | dynamic | 0.479 | 0.460 | 0.445 | 0.426 | 0.448 |
| $B_5$ | Static | 79.147 | 66.440 | 55.159 | 47.940 | 35.883 |
| | dynamic | 199.832 | 141.231 | 93.343 | 81.101 | 69.249 |
| $B_6$ | Static | 49508 | 48858 | 49296 | 40372 | 34552 |
| | dynamic | 43943 | 25126 | 32766 | 21907 | 37047 |
| $B_7$ | Static | 0.931 | 0.863 | 0.751 | 0.679 | 0.820 |
| | dynamic | 0.805 | 0.821 | 0.805 | 0.813 | 0.759 |
| $B_8$ | Static | 1.690 | 1.630 | 1.544 | 1.768 | 1.309 |
| | dynamic | 2.159 | 2.033 | 1.885 | 2.452 | 1.619 |
| $B_9$ | Static | −0.690 | −0.630 | −0.544 | −0.768 | −0.309 |
| | dynamic | −1.159 | −1.033 | −0.885 | −1.452 | −0.619 |
| $B_{10}$ | Static | 1.552 | 1.515 | 1.389 | 1.603 | 1.209 |
| | dynamic | 1.933 | 1.848 | 1.712 | 2.181 | 1.470 |
| $B_{11}$ | Static | −0.552 | −0.515 | −0.389 | −0.603 | −0.209 |
| | dynamic | −0.919 | −1.120 | −0.724 | −0.978 | −0.576 |

It can be seen that when the static load changed to a dynamic load, the brittleness value calculated by brittleness indices $B_4$, $B_6$, and $B_7$ weakened instead. $B_4$ was determined by the stress reduction degree and stress reduction rate at the post−peak stage of the stress–strain curve. However, in the SHPB test, the residual strength was generally zero and the correctness of the post−peak curve is still controversial. Therefore, $B_4$, which determines rock brittleness only through the post−peak brittleness characteristics, was not suitable for measuring rock brittleness under dynamic load. $B_6$ had the same problem. $B_7$ was determined based on the relationship between elastic strain energy and total energy before peak. Only the energy variation characteristics of the pre−peak stage were considered, and the effects of compressive strength and elastic modulus were ignored, which resulted in incorrect results.

The brittleness of the rock samples was reflected not only in the pre−peak stage, but also in the post−peak stage. In the pre−peak stage, the more brittle the rock, the larger the elastic modulus and the smaller the unrecoverable strain. In the post−peak stage, the more brittle the rock, the greater the reduction in stress and the faster the reduction rate. Since $B_8$ and $B_{10}$ take into account pre−peak and post−peak brittleness characteristics and are more comprehensive than other indices, the subsequent studies in this section will mainly use $B_8$ and $B_{10}$ to explore the brittleness response characteristics of rocks.

*3.3. Brittleness Response Characteristics under Different Loading Modes*

As can be seen from Table 3, when the load on the rock changed from static to dynamic, the brittleness of the rock was enhanced, while the relative brittleness relationships

between different rocks did not change. A rock with stronger brittleness under static load must be stronger than other rocks under dynamic load, with a "strong and constant strength" feature.

In this paper, the difference in brittleness index values of rock samples under dynamic load and static load is defined as brittleness value difference. Figures 6 and 7 show the relationship between the brittleness value difference and the static brittleness value. Each black dot represents a type of rock. It can be seen that the greater the brittleness value under static load, the greater the brittleness value difference. In other words, for a rock with strong brittleness under static load, its brittleness value was increased more significantly under dynamic load. This also results in the relative brittleness relationship between different rocks not changing according to the loading method.

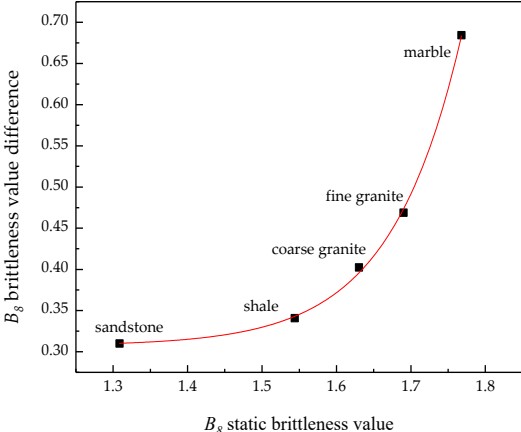

**Figure 6.** The relationship between the brittleness value difference and the static brittleness value based on $B_8$.

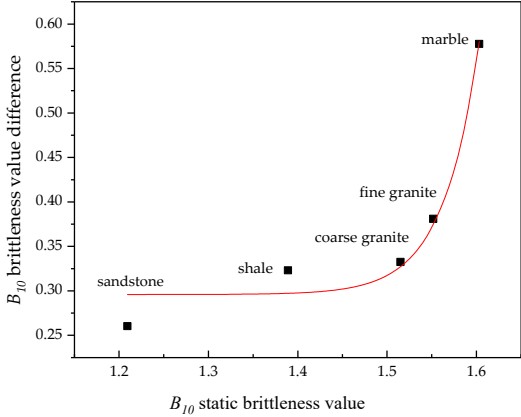

**Figure 7.** The relationship between the brittleness value difference and the static brittleness value based on $B_{10}$.

### 3.4. Changes of Rock Brittleness Indices at Different Strain Rates

Figure 8 shows the dynamic stress–strain curves of sandstone under different strain rates. It can be seen that with the increase in strain rate, the elastic deformation became larger and larger, showing an obvious strain rate effect. At the same time, it can be seen that the degree of plastic deformation was affected by the strain rate and tended to decrease with the increase in strain rate. The smaller the pre−peak plastic deformation was, the less energy was dissipated and the more elastic energy was stored, thus resulting in a greater possibility of brittle failure. This shows that the brittleness of rock increases with an increasing strain rate.

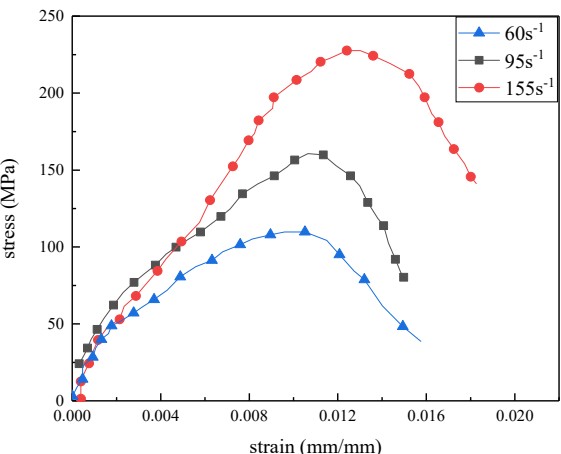

**Figure 8.** Dynamic stress–strain curves of sandstone at different strain rates.

Figure 9 shows the changes in some brittleness indices under different strain rates. It can be seen that, with the increase in strain rate, the brittleness of rock generally showed an increasing trend. Taking brittleness index $B_8$ as an example, as the strain rate increased from 60 s$^{-1}$ to 95 s$^{-1}$, the brittleness index value increased from 1.50 to 1.57, an increase of 0.07; however, when the strain rate increased from 95 s$^{-1}$ to 155 s$^{-1}$, the brittleness index value increased from 1.57 to 1.80, an increase of 0.23. $B_3$, $B_6$, $B_9$, $B_{10}$, and $B_{11}$ all have the same rule. This indicates that with the increase in strain rate, the greater the degree of brittleness enhancement, the more obvious the brittleness of the specimen becomes, and the more intense the failure process is.

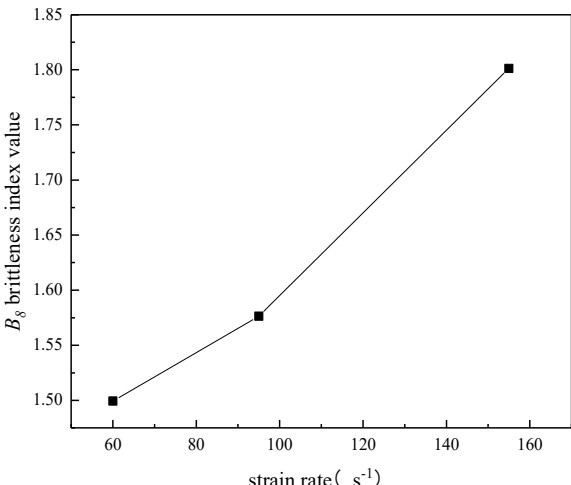

**Figure 9.** Changes in $B_8$ at different strain rates.

## 4. Discussion

### 4.1. Narrow Brittleness Definition and Generalized Brittleness Definition

The definition of brittleness was initially aimed at isotropic homogeneous materials such as metals and ceramics. It is generally believed that metals have strong toughness, while ceramics and glass have strong brittleness. However, rock materials have obvious anisotropy and inhomogeneity, so the definition of brittleness that is directly applied to metal and ceramic is controversial. Some scholars [38–40] believe that brittleness is an inherent attribute of rocks which is only related to mineral composition and structural composition, and the influence of environmental and stress state changes on rock brittleness should not be considered. Another group of scholars [41–44] believes that different external factors (such as water content, temperature, confining pressure, etc.) will inevitably lead to changes in the brittleness states and failure modes of rock materials. When evaluating

brittleness changes in rock, internal and external conditions such as temperature, stress state, and microstructure should be comprehensively considered.

It can be seen from the tests carried out in this paper that, under different loading modes and different strain rates, the brittleness values calculated by all of the brittleness indices underwent great changes. At the same time, various characteristics considered to reflect the brittleness of rock, such as compressive strength and elastic modulus, also underwent significant changes, indicating that the brittleness of rock will change with different loading rates. However, all changes were based on an initial state, so this paper posits that rock brittleness can be defined in two ways. One is the narrow sense of rock brittleness, and the other is the broad sense of rock brittleness. The narrow brittleness definition classifies it as an inherent property of rock materials, i.e., a mechanical behavior characteristic of rock under natural conditions. The generalized brittleness definition consists of two parts: one is the brittleness of rock foundations, and the other is the increase or decrease in the brittleness of rocks. Rock foundation brittleness is equivalent to the narrow rock brittleness definition. The increase or decrease in the brittleness of rocks is the change in rock brittleness under different influencing factors (loading rate, confining pressure, temperature, water content, etc.).

### 4.2. Changes in Brittleness Characteristics of Rock under Dynamic Load

The total stress–strain curve obtained from the rock compression test is a direct reflection of the mechanical behavior of rock, and can reveal the internal mechanism of rock failure. Therefore, the total stress–strain curve is often used as a qualitative and quantitative method to evaluate the brittleness of rock. Based on the summary of the existing brittleness indices and the previous experimental study, this section summarizes the rock brittleness in terms of the stress–strain curve in four categories.

The first brittleness characteristic is the strain variation in the stress–strain curve, which is mainly represented by the relationship between the plastic strain and the peak strain. The more brittle the rock, the smaller the ratio of the plastic strain to the peak strain. The second brittleness characteristic is the stress variation in the stress–strain curve, which is mainly manifested by the change in peak stress amplitude. The more brittle the rock, the greater the peak strength and the greater the post−peak stress drop. The third brittleness characteristic is the stress–strain relationship in the stress–strain curve, which is mainly represented by the pre−peak elastic modulus and the post−peak elastic modulus. The more brittle the rock, the larger the elastic modulus and the stronger the resistance to deformation. At the same time, the stress drop rate is faster and the post−peak elastic modulus is larger. The fourth brittleness characteristic is the energy relationship in the stress–strain curve, which is mainly manifested as the energy absorption or release of the rock after failure. The more brittle the rock is, the less energy it can absorb after reaching the peak, and even self−sustaining failure can occur without absorbing energy.

According to the test results, when a static load changes to a dynamic load, the plastic strain ratio of the rock decreases, the peak strength of the rock increases significantly, the pre−peak elastic modulus of the rock increases to varying degrees, the energy released after the rock's failure is greater, and the four types of brittleness characteristics are enhanced, so that the overall brittleness of the rock is stronger.

In Section 3, it was described that the brittleness of rock was enhanced to different degrees under dynamic load, and the enhancement amplitude was positively correlated with the brittleness of rock under static load. At different strain rates, the brittleness of sandstone increases with the increase in strain rate, and the higher the strain rate, the greater the increase in brittleness. This is mainly due to the loading rate, which amplifies the changes in brittleness characteristics. The energy absorbed by the rock is used to compress itself and spread the fracture before failure. Under static load, the loading rate is very low, all the energy can be fully absorbed, and fracture propagation in the rock is only related to the mineral composition and structural composition. However, under dynamic load, the loading rate is relatively high, and not all of the energy can be fully absorbed by

the rock to be compressed in a short period of time. Part of this energy is used to accelerate the expansion of cracks, resulting in more obvious brittleness characteristics.

## 5. Conclusions

In this paper, a series of dynamic load tests and static load tests were carried out on five different types of rocks. The brittleness changes in rock under dynamic load and in sandstone under different strain rates were studied. It was also discussed whether the existing brittleness index could be used to describe the changes in the brittleness of the rock samples under dynamic load. The main research conclusions are as follows:

(1) Under the influence of different loading rates, the brittle state and failure mode of the rock changed significantly. This paper considers that rock brittleness can be defined in two ways: one is the narrow brittleness definition, and the other is the generalized brittleness definition. The narrow brittleness definition is an inherent property of rock material, characteristic of its mechanical behavior. The generalized brittleness definition includes the narrow sense of brittleness and the degree of increase in brittleness according to different influencing factors (loading rate, confining pressure, temperature, water content, etc.)

(2) Loading rate amplifies changes in the brittleness characteristics of rock. When static load changes to dynamic load, the brittleness of the rock is enhanced to different degrees, and the enhancement amplitude is positively correlated with the brittleness of the rock under static load. The brittleness of sandstone also has an obvious strain rate effect. The brittleness of rock increases with the increase in strain rate, and the greater the strain rate, the greater the degree of brittleness enhancement.

(3) Brittleness indices $B_8 \sim B_{11}$, based on the energy relationship, take into account the brittleness failure characteristics of a rock in both the pre−peak stage and the post−peak stage at the same time. It can accurately describe the brittleness changes in rocks under different loading conditions and different strain rates, and is suitable for measuring the brittleness of rocks under dynamic load.

**Author Contributions:** Conceptualization, D.L. and M.H.; Methodology, M.H.; Validation, D.L.; Investigation, M.H. and Q.Z.; Resources, D.L. and Q.Z.; Data curation, M.H.; Writing—original draft, M.H.; Writing—review & editing, D.L. and Q.Z.; Visualization, M.H.; Supervision, Q.Z.; Project administration, D.L. and Q.Z.; Funding acquisition, D.L. All authors have read and agreed to the published version of the manuscript.

**Funding:** This research was funded by National Natural Science Foundation of China, grant number No. 52074349.

**Institutional Review Board Statement:** Not applicable.

**Informed Consent Statement:** Not applicable.

**Data Availability Statement:** The data used to support the findings of this study are available from the corresponding author upon request.

**Conflicts of Interest:** The authors declare no conflict of interest.

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
