# Peer review of "Evaluation of Rock Brittleness Index under Dynamic Load"

_applsci, doi:10.3390/app13084698_

Round 1
Reviewer 1 Report
The paper is generally well written and has a logical arrangement, however I have a few questions and comments:
1. In the chapter "materials and methods" the Authors write about the test methods, the equipment on which the tests were carried out. Also, the method of analysing of SHPB test data were described in great detail.
However, the method for processing the results of uniaxial compression and the Brazilian test was omitted. Please add necessary information even in short form.
2. Strains were measured using the electroresistive method. What type of strain gauges were used and to what equipment were they connected during both the static and dynamic (SHPB) tests.
3. Each type of test was carried out on three samples, which is a very small number in the case of rock. What are the mean values of the results and their standard deviations. What do the data in the tables and graphs represent (for all manuscript). Are these average values or maybe it is the results for a specific sample? In the case of rock material, the level of repeatability of the recorded data is very important.
4. What do the points (black squares) in Figures 4 and 5 represent?
5. How did the data shown in Figure 6 obtained.
Please respond to the above comments.
Author Response
Thank you very much for kindly processing our manuscript and providing us valuable comments. We appreciate your comments and suggestions to improve the quality of our manuscript. We have tried our best to revise the manuscript based on the comments and suggestions. The revision part is highlighted by red fonts in the text. A detailed reply to your comments and suggestions is attached.

Reviewer 2 Report
I have gone through an article entitled " Evaluation of rock brittleness index under dynamic load" where authors tried to check rock brittleness by different rock static and dynamic testing on five different rock types. Three types of tests were conducted static uniaxial compression test, Brazilian splitting test, and dynamic compression test. Authors found that dynamic tests have high brittleness as compared to static. The article's English is fine and the citing references are also well. Failure of rock in the different modes under loading is interesting and required deep discussion up to the micro-scale level of rock materials. The flow of the paper presentation is well written but the results need more justifications, so there are some shortcomings, which need to be addressed:
Authors should justify why shear failure is the main failure mode of fine granite, shale, marble, and sandstone, while split failure is the main failure mode of coarse granite and shale. Under dynamic load, the failure modes of the five kinds of rock are mainly split failure. The five kinds of rocks show brittle failure characteristics under different loading modes. Also, provide appropriate references for the above justification.
Table 2 shows the strain rate but seems confusing that it have the same strain rate on static and dynamic testing, whereas fig 2 mentioned there were different strain rate on different types of testing, both are contradictory to your statements. Please check it.
Authors should define different types of failure modes in one paragraph in the literature review with figures…
Comment on line 175- Rock failure is a serious problem in rock engineering environments. Rock failure modes, however, are complex and difficult to quantify or predict. A comprehensive study on rock failure modes at the laboratory scale is, therefore, potentially important as it helps recognize the adequacy of the support designed on the basis of the nature of engineering work. The nature of the principal failure mode changes from axial splitting to shearing along a single plane to multiple fracturing in the case of both granite and sandstone specimens as uniaxial compressive strength (UCS) increases. The relation between failure modes of granite and sandstone rocks under uniaxial compression and corresponding UCS values was broadly explained in terms of the damage evolution of the rocks. Granite and sandstone specimens failed mainly following central or central multiple types of fracturing. In the case of granite and sandstone, the central multiple failure modes corresponds to high tensile strength. It was found that granite and sandstone specimens generally fail through the rock materials in one or more extensional planes containing the line of loading.
Kindly go through the above suggestion for different modes of failure and modified your statements also. Authors should provide more supportive outcomes of why these happened, if possible, i.e., photomicrograph of rock sample before the test, during the test, and after test, etc.
The introduction should have more references related to work done by other researchers. Some of the references are cited but not included in the references list and vice versa. There are references suggested for your article, it might help to strengthen authors' findings, which are:- Correlation of ultrasound velocity with physico-mechanical properties of Jodhpur sandstone (for application of static load), A Numerical Simulation of Influence of Rock Class on Blast Performance (for application of dynamic load), Digital rock physics and application of high-resolution micro-CT techniques for geomaterials (for the understanding of rock materials up to micro-scale level)
Author Response

(The authors gave the same response as above.)
